# Prospective Evaluation of Quality of Life and Functional Outcomes after Carbon Ion Radiotherapy for Inoperable Bone and Soft Tissue Sarcomas

**DOI:** 10.3390/cancers13112591

**Published:** 2021-05-25

**Authors:** Shuichiro Komatsu, Masahiko Okamoto, Shintaro Shiba, Takuya Kaminuma, Shohei Okazaki, Hiroki Kiyohara, Takashi Yanagawa, Takashi Nakano, Tatsuya Ohno

**Affiliations:** 1Department of Radiation Oncology, Gunma University Graduate School of Medicine, 3-39-15, Showa-machi, Maebashi 371-8511, Gunma, Japan; m1720022@gunma-u.ac.jp (S.K.); shiba48@gunma-u.ac.jp (S.S.); cami_taku@gunma-u.ac.jp (T.K.); s_okazaki@gunma-u.ac.jp (S.O.); tnakano@gunma-u.ac.jp (T.N.); tohno@gunma-u.ac.jp (T.O.); 2Gunma University Heavy Ion Medical Center, 3-39-15, Showa-machi, Maebashi 371-8511, Gunma, Japan; 3Department of Radiation Oncology, Japanese Red Cross Maebashi Hospital, 389-1, Asakura-machi, Maebashi 371-0811, Gunma, Japan; hiroki.kiyohara@maebashi.jrc.or.jp; 4Department of Orthopedic Surgery, Gunma University Hospital, 3-39-15, Showa-machi, Maebashi 371-8511, Gunma, Japan; tyanagaw@gunma-u.ac.jp; 5National Institute of Radiological Sciences, National Institutes for Quantum and Radiological Science and Technology, 4-9-1, Anagawa, Inage-ku, Chiba-shi 363-8555, Chiba, Japan

**Keywords:** inoperable bone and soft tissue sarcoma, carbon ion radiotherapy, quality of life, functional outcome

## Abstract

**Simple Summary:**

Quality of life (QOL) and functional outcomes in patients with inoperable bone and soft tissue sarcoma treated with definitive carbon-ion radiotherapy (CIRT) were prospectively investigated. CIRT showed favorable clinical efficacy and safety, maintaining the physical component of QOL and functional outcomes, and improving the mental component of QOL. The physical component of QOL was positively correlated with functional outcomes. Poor performance status at diagnosis and female gender were independent predictors of the physical component of QOL and functional outcomes after CIRT.

**Abstract:**

Carbon-ion radiotherapy (CIRT) represents a definitive treatment for inoperable bone and soft tissue sarcoma (BSTS). This prospective study analyzed 61 patients with inoperable BSTS who were treated with CIRT to evaluate QOL, functional outcomes, and predictive factors in patients with inoperable BSTS treated with definitive CIRT. The Musculoskeletal Tumor Society (MSTS) scoring system and the Short Form (SF)-8 questionnaire were completed before and at 1, 3, 6, 12, and 24 months after CIRT. The median follow-up period was 38 months. The main site of primary disease was the pelvis (70.5%), and the most common pathologic diagnosis was chordoma (45.9%). The 3-year overall survival and local control rates were 87.8% and 83.8%, respectively. The MSTS score and physical component score (PCS) of SF-8 did not change significantly between the baseline and subsequent values. The mental component score of SF-8 significantly improved after CIRT. Multivariate analysis showed that the normalized MSTS and normalized PCS of SF-8 at the final follow-up were significantly affected by performance status at diagnosis and sex. CIRT showed clinical efficacy, preserving the physical component of QOL and functional outcomes and improving the mental component of QOL, suggesting its potential value for the treatment of patients with inoperable BSTS.

## 1. Introduction

Bone and soft tissue sarcoma (BSTS) is a rare disease and one of the most difficult tumors to cure [1,2]. Although the standard of care for patients with BSTS is complete tumor resection, a standard treatment for inoperable BSTS remains to be established. The primary treatment approaches for inoperable BSTS incorporate chemotherapy and photon therapy [3,4]. However, these strategies show limited therapeutic efficacy [3,4]. Uncontrolled BSTS decreases physical function [5], which leads to a lower performance status in patients [6]. Recent studies on carbon ion radiotherapy (CIRT) and proton radiotherapy have shown clinical efficacy for the treatment of inoperable BSTS [7,8,9,10,11,12,13]. CIRT, an innovative radiotherapy modality that achieves high dose conformity to deeply located tumors, is more effective than photon therapy [14]. In patients with inoperable BSTS, CIRT shows beneficial therapeutic outcomes, resulting in 5-year local control (LC) rates of 65–79% and 5-year overall survival (OS) rates of 46–81.1% [7,8,9,10]. Thus, CIRT is recently recognized as a choice for the treatment of inoperable BSTS [11].

In curative cases, quality of life (QOL) plays an important role in improving OS in the field of the treatment of patients with other malignant tumors [15,16]. Therefore, the maintenance or improvement of QOL needs to be considered when choosing a treatment option. The effects of surgery on QOL and functional outcomes in patients with resectable BSTS have been investigated extensively [17,18]. However, few studies have investigated QOL and functional outcomes in patients with inoperable BSTS treated with chemotherapy and/or photon radiotherapy, and the effect of CIRT on QOL and functional outcomes in this patient population remains unclear.

In this study, we prospectively analyzed the effect of definitive CIRT on QOL and functional outcomes, as well as the predictive factors and correlation between QOL and functional outcomes in patients with inoperable BSTS.

## 2. Materials and Methods

### 2.1. Patient Selection

This study was a prospective observational study that analyzed patients with pathologically diagnosed inoperable BSTS who were treated with CIRT between April 2011 and March 2017. All tumors were staged according to The Cancer Staging Manual of the American Joint Committee on Cancer 8th edition classification. Decisions on tumor resectability and inoperability were made by a cancer review board that included orthopedic surgeons, diagnostic radiologists, radiation oncologists, medical oncologists, and pathologists at our hospital. Patients with inoperable BSTS in our study included elderly patients with impaired surgical tolerance, patients with too serious complications to receive surgery, patients who refused surgery, and patients who had locally advanced BSTS with or without resistance to chemotherapy. The patients met all of the following eligibility criteria: no nodal or distant metastasis, Karnofsky performance status (KPS) between 60 and 100, no intravascular tumor embolism, site other than the head and neck, no invasion to the gastrointestinal tract, no infection at the tumor site, and few severe complications with expected survival of >6 months. A minimum follow-up period of 1 year was required for assessing QOL and functional outcomes. The observational period of >1 year was consistent with most previous studies [19]. The exclusion criteria were as follows: history of irradiation to the same site of primary disease and active double cancers. Late adverse events were graded using the Common Terminology Criteria for Adverse Events, version 4.0 of the National Cancer Institute at the end of the 90-day period after CIRT. Patients were treated according to our protocols registered with the University Medical Information Network (UMIN, Clinical Trial Registry number 000009720). This study was approved by the Institutional Review Board of our institution (approval number: 765) and was conducted following the local Ethics Committee guidelines.

### 2.2. Carbon Ion Radiotherapy

CIRT was performed with definitive intent in all patients. The gross tumor volume (GTV) was delineated with reference to contrast-enhanced computed tomography/magnetic resonance imaging and 18-fluorodeoxyglucose positron emission tomography data. The clinical target volume (CTV) included the potential area of tumor spread and the planning target volume (PTV), as well as an additional 3 mm margin from the CTV. CTV and PTV margins were modified as necessary when the targets were close to organs at risk. PTV was modified not to be exceed outside the patient. The radiation dose was prescribed at the isocenter of the PTV. The PTV encompassed the 95% isodose line of the prescribed dose. The CIRT dose was expressed as Gy relative biological effectiveness (RBE) set at 3.0 based on previous experimental data [20]. The usual dose was 70.4 Gy (RBE) administered in 16 fractions; however, 67.2 Gy (RBE) and 64 Gy (RBE) were administered in 16 fractions for sacral chordoma and spinal/paraspinal sarcoma, respectively. The dose constraints of OARs were defined as follows: a maximum dose of 30 Gy (RBE) for the spinal cord, and 60 Gy (RBE) <20 cm2 for the skin.

### 2.3. Functional Outcome and Quality of Life Assessment

The Musculoskeletal Tumor Society (MSTS) scoring system [21,22,23] was used to assess functional outcomes, and the Short Form (SF)-8 questionnaire [24,25] was used to evaluate health-related QOL before/after CIRT. The MSTS score was assigned by the attending physician and used to measure anatomical function, whereas SF-8 questionnaires were completed by patients to document physical disability degree and general health status. The SF-8 questionnaires were administered by medical staff other than the physician to avoid interviewer bias. The MSTS scoring system is a physician-reported scale for assessing functional outcomes and is one of the most commonly used questionnaires in this field [19]. The MSTS score is based on six items: pain, overall function, and emotional acceptance, and three items specific to upper or lower body tumors. Each item is rated on a scale of 0–5, where 5 is the most favorable score. The total score is calculated by adding the individual items on a scale of 0–30. The SF-8 questionnaire includes eight domains that are summarized by the physical component score (PCS) and mental component score (MCS). Higher numerical scores indicate better QOL. The MSTS evaluation and SF-8 questionnaire were administered before (baseline) and at 1, 3, 6, 12, and 24 months after CIRT. The score at the time of follow-up was normalized by dividing it by the baseline score before CIRT. All patients were not indicated to receive specific rehabilitation therapies.

### 2.4. Statistical Analysis

OS was defined as the interval between the start of CIRT and the date of the last follow-up or death. LC was defined as the absence of local recurrence or recurrence in the original anatomical site of involvement. Differences in sequential changes of the normalized MSTS score and normalized QOL scores (PCS and MCS of SF-8) were evaluated using the Friedman test. When there were significant differences, differences in the normalized scores at every time point were evaluated using the Wilcoxon signed-rank test with Bonferroni correction. Spearman rank correlation analyses were used to evaluate the relationships between the total scores in the scales (MSTS score, and PCS and MCS of SF-8) and the relationships between KPS at diagnosis and each total score (MSTS score, and PCS and MCS of SF-8) at diagnosis. To identify independent predictors of the normalized score at the final follow-up, uni- and multivariate analyses were performed using a linear regression model. The final follow-up point was 12 months or 24 months after CIRT, whichever was available. A multivariable logistic regression model was generated using a set of covariates based on clinical relevance, including local recurrence, late adverse event, GTV, KPS, sex, and total dose.

All statistical analyses were performed using R 3.5.2 (The R Foundation for Statistical Computing, Vienna, Austria). A *p*-value of <0.05 was considered statistically significant.

## 3. Results

This section may be divided by subheadings. It should provide a concise and precise description of the experimental results, their interpretation, as well as the experimental conclusions that can be drawn.

### 3.1. Patient Characteristics

The present cohort included 61 patients. The median follow-up period was 38 months (range, 12.4–102.6 months). The patient characteristics (Table 1) indicate that the cohort was composed of a variety of patients with inoperable BSTS. The histopathological type was chordoma in almost half of the patients (45.9%). The predominant primary tumor site was the pelvis (70.5%). Primary disease was located in the trunk in 51 patients (83.6%), and 44 patients (72.1%) had a primary tumor larger than 8 cm.

### 3.2. Clinical Efficacy and Safety

The efficacy of CIRT was evaluated by assessing the clinical outcomes of the cohort. The 3-year OS and LC rates were 87.8% [95% confidence interval (CI), 81.6–97.2%] and 83.8% (95% CI, 68.1–92.2%), respectively (Figure 1A,B). Late adverse events are shown in Appendix A. The most common late adverse events were dermatitis and peripheral sensory neuropathy. There were no late adverse events of grade 3 or higher except for grade 3 osteomyelitis in one patient.

### 3.3. QOL and Functional Outcomes Following CIRT

The MSTS score and QOL scores (PCS and MCS of SF-8) were used to assess the effect of CIRT on QOL and functional outcomes in patients with inoperable BSTS. The normalized MSTS score and normalized PCS and MCS of SF-8 are shown in Figure 2A–C. The median baseline MSTS score, PCS, and MCS of SF-8 were 20.00 [interquartile range (IQR), 14.75–24.25], 41.04 (IQR, 36.44–49.49), and 43.75 (IQR, 36.91–48.08), respectively. The normalized MSTS score and normalized PCS of SF-8 did not change significantly between the baseline and subsequent measurements (*p* = 0.26 and 0.085, respectively, Figure 2A,B). The normalized MCS of SF-8 improved significantly after completion of treatment throughout the observation period (*p* < 0.001, Figure 2C).

The relationships among the total scores of the scales (MSTS and PCS and MCS of SF-8) were evaluated by analyzing the correlation between functional outcomes and QOL scores (Figure 3A–C). KPS, PCS and MCS of SF-8 were determined independently in our study. The MSTS score was positively correlated with the PCS of SF-8 (Figure 3A; correlation coefficient: 0.55; *p* < 0.001). A weak positive correlation was detected between the MSTS score and the MCS of SF-8 (Figure 3B; correlation coefficient: 0.25; *p* < 0.01). A similar but weaker positive correlation was observed between the PCS and MCS of SF-8 (Figure 3C; correlation coefficient: 0.11; *p* = 0.03).

### 3.4. Identification of Predictors of Functional Outcomes and QOL by Uni- and Multivariate Analyses

Uni- and multivariate analyses were performed to identify independent predictors of the functional outcome and QOL scores at the final follow-up after normalizing to the baseline. The results of univariate analyses (Table 2) showed that sex, KPS, and total dose were significant predictive factors for the normalized MSTS score and PCS of SF-8 at the final follow-up. Female gender, poor baseline KPS, and a high total dose contributed to a favorable normalized MSTS score and normalized PCS of SF-8 at the final follow-up.

The results of multivariate analyses using a linear regression model (Table 3) indicated that the normalized MSTS and normalized PCS of SF-8 at the final follow-up were significantly affected by both KPS at diagnosis (*p* < 0.0001 and 0.0037, respectively) and sex (*p* = 0.0052 and 0.045, respectively). Female patients and those with worse KPS at diagnosis showed a significant improvement of the MSTS and PCS of SF-8 at the final follow-up.

To determine whether the impairment of functional outcomes and QOL was associated with a poor performance status caused by the primary tumor, the correlations between KPS at diagnosis and the baseline value of each score (MSTS and SF-8 PCS and MCS) were investigated. The results are summarized in Figure 4A–C. KPS at diagnosis showed a significant positive correlation with the baseline MSTS score (Figure 4A; correlation coefficient: 0.66; *p* < 0.0001) and PCS of SF-8 (Figure 4B; correlation coefficient: 0.48; *p* < 0.0001). KPS at diagnosis was not significantly correlated with the MCS of SF-8 (Figure 4C; correlation coefficient: −0.087; *p* = 0.49).

## 4. Discussion

This study firstly investigated the QOL and functional outcomes of 61 BSTS patients who were followed up for >1 year after CIRT. Although previous studies used the MSTS scoring system to analyze the functional outcomes of five to seven patients before and just after CIRT, they did not assess QOL by using tools such as the SF-8 questionnaire [10,26]. In contrast, both QOL and functional outcomes were measured before and at 1, 3, 6, 12, and 24 months after CIRT in our study. We demonstrated the most comprehensive and long-term results on QOL and functional outcomes in patients with inoperable BSTS receiving CIRT, and thus, our study could provide important information to determine a choice of treatment in patients with inoperable BSTS.

We demonstrated the clinical efficacy of CIRT for the treatment of inoperable BSTS. The 3-year OS and LC rates were 87.8% and 83.8%, respectively. There was no grade 3 or higher late adverse events including neuropathy, except for one patient with grade 3 osteomyelitis (Appendix A). The clinical outcomes in our cohort were comparable to those of previous studies reporting 3-year OS 59–92% and 3-year LC rates of 68–84% on CIRT for inoperable BSTS [7,8,9,10]. The most common histopathological type was chordoma (45.9%), and the most prevalent anatomical site of disease was the trunk, including the sacrum, pelvis, chest, and retroperitoneum in our study (Table 1). The distribution of treatment sites was mainly derived from the selection of patients who are likely to be inoperable [27].

The present study demonstrated not only that the MSTS and PCS of SF-8 were maintained, but also that the MCS of SF-8 was significantly improved even at 12 and 24 months after CIRT (Figure 2A–C). A similar trend of improved mental component of QOL after treatment was reported in a prospective study of CIRT for locally advanced head and neck cancers and a retrospective study of proton radiotherapy for rhabdomyosarcoma in children [13,28]. In addition, definitive surgery for resectable BSTS also improved mental health status and maintained physical/functional health status [29,30]. The population of surgical studies consisted of extremity of soft tissue sarcoma; therefore, differences in patient characteristics between the present and previous studies need to be considered. The MSTS and QOL reported in some previous studies were summarized in Appendix A).

Interestingly, there was a significantly positive correlation between the MSTS and PCS of SF-8 (correlation coefficient: 0.55; *p* < 0.001; Figure 3). The MSTS score assesses the anatomical function as reported by the physician, whereas the PCS of SF-8 measures physical disability as reported by the patient. These two indexes were independently determined in our study. Jansen et al. similarly demonstrated that the Patient-Reported Outcomes Measurement Information System funded by the National Institute of Health was significantly positively correlated to the MSTS score reported by physicians in patients with lower extremity bone metastases [31]. Since our study included various sites of primary tumor, further studies are necessary to investigate the impact of tumor location on the correlation between the PCS of SF-8 and MSTS scores in inoperable BSTS treated with CIRT.

The independent predictive factors for QOL and functional outcomes after CIRT were identified using uni- and multivariate analyses (Table 2 and Table 3) and correlation analyses (Figure 4A–C). Patients with a poor KPS before CIRT and female patients in our cohort showed a significant improvement in the MSTS (*p* < 0.0001 and 0.0085, respectively) and PCS of SF-8 (*p* < 0.001 and 0.014, respectively) after CIRT. The MSTS and PCS of SF-8 (associated with the primary tumor) in BSTS were significantly positively correlated with performance status before CIRT (correlation coefficient: 0.66; *p* < 0.0001 and correlation coefficient: 0.48; *p* < 0.0001, respectively). A poor performance status before CIRT in this study cohort was largely caused by the primary BSTS. Local tumor control by CIRT improved the MSTS score and PCS of SF-8, which may lead to an improved KPS after CIRT. The present results thus suggest that patients with a poor KPS had substantial room for improvement of the KPS after CIRT by improving QOL and functional outcomes in the best scenario. On the other hand, size of GTV did not affect functional outcomes and the PCS of SF-8 in this study cohort, whereas only radiation field size corresponding to GTV was a significant functional prognostic factor for BSTS patients receiving pre-operative photon therapy [32,33]. The difference may be attributed to the fact that CIRT has an excellent dose conformity to the target while minimizing the damage to the surrounding organs at risk (i.e., muscle, bone, and joints) [34]. We did not identify factors affecting the MCS of SF-8 in this study. However, there is a possibility of unmeasured factors affecting mental conditions before and after CIRT.

Chemotherapy and photon therapy are considered first-line treatments for inoperable BSTS [3,4]. The primary aim of chemotherapy in patients with inoperable BSTS is to delay disease progression and maintain QOL for as long as possible [35]. Several publications reported that QOL scores decline during chemotherapy because of its adverse effects, such as diarrhea, anorexia, nausea, vomiting, and fatigue [36,37]. There are no studies investigating the effect of chemotherapy and photon radiotherapy on functional outcomes in patients with inoperable BSTS. By contrast, CIRT for inoperable BSTS as definitive local therapy achieved “favorable” clinical outcomes and improved QOL in our study. Recent studies reported that higher QOL was even associated with prolonged OS in the field of the treatment of patients with other malignant tumors [15,16]. The present results suggest that CIRT is an important treatment option for inoperable BSTS in terms of preserved functional outcomes and improved QOL.

Patients with inoperable BSTS in our cohort included elderly patients with impaired surgical tolerance, patients with too serious complications to receive surgery, patients who refused surgery to avoid complications from surgery, and patients who had locallyadvanced BSTS with or without resistance to chemotherapy. The decision on the indication for CIRT was made on a risk/benefit consideration by a cancer review board that included orthopedic surgeons, diagnostic radiologists, radiation oncologists, medical oncologists, and pathologists. CIRT for inoperable BSTS has been covered by public insurance in Japan because of its beneficial therapeutic outcomes [7,8,9,10]. However, CIRT is currently an extremely limited medical resource, with fewer than twenty facilities in the world [38,39]. We should consider the potential bias deriving from decisions on inoperability in the individual situation with different medical options.

This study had several limitations. First, whether the results can be extended to the general population of patients with BSTS is questionable because the present study cohort included mainly patients with BSTS in the trunk and sacral chordomas. Second, because of the heterogeneous features of BSTS, we did not evaluate the functional outcomes and QOL according to the anatomic sites and histopathologic types. We should also consider the potential source of bias deriving from a single-institutional study. Further study with a larger cohort with longer follow-up is necessary to confirm the results. Third, we used SF-8 questionaires to access QOL of the patients in our study. The SF-8 has been shown to be effective monitoring population health and large-scale outcomes studies, while the confidence of SF-8 employed in relatively small number of patients suffering from rare diseases has not been established [40]. Our findings should be validated by different scales other than SF-8 for assessing QOL. Fourth, we did not conduct the genetic testing for targeted therapies in our study. The advent of targeted therapies may lead to an improvement of treatment options and clinical outcomes in some kinds of BSTS [41]. The cost of genetic testing, which is useful for diagnosing soft tissue tumors, was not covered by public medical insurance in our country, and there were few facilities with established genetic testing systems, so the genetic testing was not possible.

## 5. Conclusions

CIRT showed clinical efficacy and safety for the treatment of inoperable BSTS, preserving the physical component of QOL and functional outcomes, and improving the mental component of QOL. CIRT is therefore recommended as a treatment option for patients with inoperable BSTS.

## Figures and Tables

**Figure 1 cancers-13-02591-f001:**
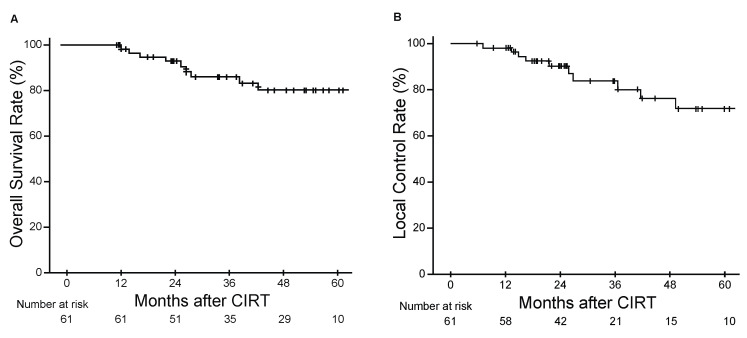
Kaplan-Meier estimates of overall survival (**A**) and local control (**B**) rates.

**Figure 2 cancers-13-02591-f002:**
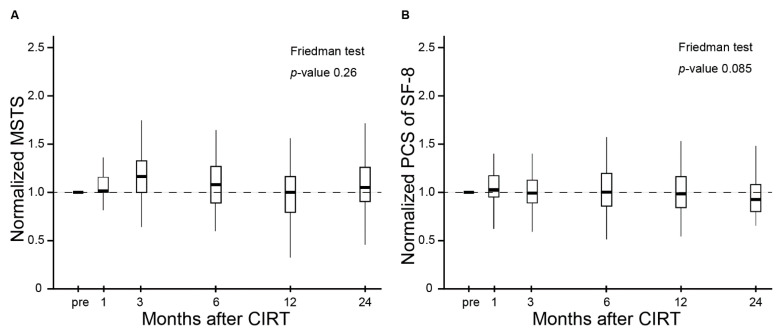
The MSTS scores, PCS of SF-8, and MCS of SF-8 are shown in box-plots including median values and interquartile ranges before CIRT and at 1, 3, 6, 12, and 24 months after CIRT. (**A**) The MSTS score did not change during the follow-up period. (**B**) The PCS of SF-8 did not change during the follow-up period. (**C**) The MSC of SF-8 improved significantly after CIRT during the follow-up period. Abbreviations: N.S = not significant; * = *p*-value < 0.05; *** = *p*-value < 0.001.

**Figure 3 cancers-13-02591-f003:**
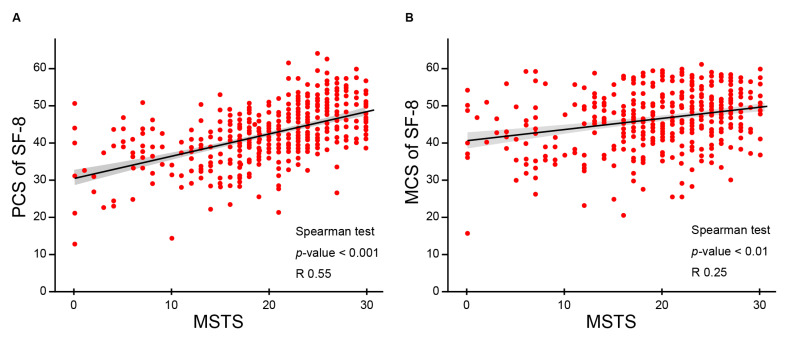
Scatter plots for MSTS scores and PCS and MCS of SF-8 are shown. Correlation estimates were generated and regression lines were drawn. The scores at all follow-up points were plotted. (**A**) MSTS scores were positively correlated with the PCS of SF-8. (**B**) MSTS scores were weakly positively correlated with the MCS of SF-8. (**C**) The PCS and MCS of SF-8 showed a weak positive correlation.

**Figure 4 cancers-13-02591-f004:**
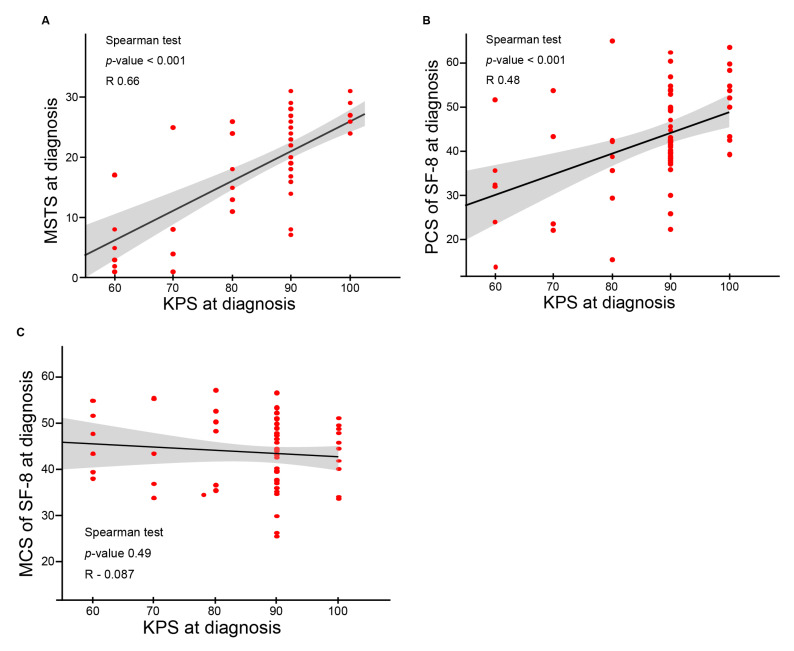
Scatter plots between KPS at diagnosis and baseline scores (MSTS scores and PCS and MCS of SF-8) are shown. Correlation estimates were generated, and regression lines were drawn. (**A**) The baseline MSTS scores were positively correlated with KPS at diagnosis. (**B**) The baseline PCS of SF-8 was positively correlated with KPS at diagnosis. (**C**) The baseline MCS of SF8 showed a weak positive correlation with KPS at diagnosis.

**Table 1 cancers-13-02591-t001:** Patient characteristics.

Characteristics	Overall, *n* = 61	Characteristics	Overall, *n* = 61
Age (mean, range)	62 (24–93)		
Gender, *n* (%)		Type, *n* (%)	
Female	19 (31.1)	Bone tumor	37 (60.7)
Male	42 (68.9)	Soft tissue tumor	24 (39.3)
KPS, *n* (%)		Histopathology, *n* (%)	
100	10 (16.4)	Chordoma	28 (45.9)
90	35 (57.4)	Chondrosarcoma	7 (11.5)
80	6 (9.8)	Liposarcoma	7 (11.5)
70	4 (6.6)	Pleomorphic sarcoma	5 (8.2)
60	6 (9.8)	Spindle cell sarcoma	4 (6.6)
Stage (Bone tumor), *n* (%)		Fibromyxoid sarcoma	3 (4.9)
Stage 1A	8 (13.1)	Sarcoma, NOS	3 (4.9)
Stage 1B	21 (34.5)	Fibrosarcoma	2 (3.3)
Stage 2A	3 (4.9)	MPNST	1 (1.6)
Stage 2B	5 (8.2)	Rhabdomyosarcoma	1 (1.6)
Stage (Soft tissue tumor), *n* (%)		Site, *n* (%)	
Stage 1A	3 (4.9)	Pelvis	43 (70.5)
Stage 1B	1 (1.6)	Limb	10 (16.4)
Stage 2	3 (4.9)	Spine/paraspine	5 (8.2)
Stage 3A	12 (19.7)	Chest	2 (3.3)
Stage 3B	5 (8.2)	Retroperitoneum	1 (1.6)
T classification (Bone tumor), *n* (%)		CIRT planning, *n* (%)	
T1	11 (18.0)	Total dose (Gy)	67.2 (64–70.4)
T2	26 (42.6)	GTV (cc)	171.80 (78.90–928)
T classification (Soft tissue tumor), *n* (%)		Tumor grading, *n* (%)	
T1	7 (11.5)	G1	28
T2	13 (21.3)	G2	12
T3	4 (6.6)	G3	15
		GX	6

Abbreviations: Sarcoma, NOS = sarcoma not otherwise specified; MPNST = malignant peripheral sheath tumor; GTV = gross tumor volume.

**Table 2 cancers-13-02591-t002:** Predictive factors for normalized scores at the final follow-up determined using univariate analyses.

Characteristics	*p*-Value			Characteristics	*p*-Value		
	MSTS	PCS	MCS		MSTS	PCS	MCS
Age	0.96	0.61	0.31	Bone or soft tissue	0.73	0.45	0.31
Sex	0.0085	0.014	0.32	Pathology			
KPS	<0.0001	<0.0001	0.39	Chordoma	0.25	0.67	0.15
Stage (Bone tumor)				Chondrosarcoma	0.96	0.39	0.57
Stage 1A	0.044	0.39	0.80	Liposarcoma	0.52	0.12	0.81
Stage 1B	0.93	0.39	0.055	Pleomorphic sarcoma	0.26	0.34	0.0018
Stage 2A	0.22	0.34	0.52	Spindle cell sarcoma	0.71	0.0013	0.16
Stage 2B	0.0017	0.089	0.34	Fibromyxoid sarcoma	0.53	0.4	0.88
Stage (Soft tissue tumor)				Sarcoma, NOS	0.0023	0.19	0.78
Stage 1A	0.84	0.89	0.15	Fibrosarcoma	0.52	0.35	0.079
Stage 1B	0.84	0.19	0.55	MPNST	0.9	0.53	0.31
Stage 2	0.66	0.081	0.67	Rhabdomyosarcoma	0.86	0.8	0.35
Stage 3A	0.50	0.49	0.70	Site			
Stage 3B	0.18	0.35	0.26	Pelvis	0.74	0.29	0.32
T classification (Bone tumor)				Limb	0.21	0.95	0.07
T1	0.26	0.16	0.58	Spine/paraspine	0.56	0.61	0.98
T2	0.11	0.071	0.15	Chest	0.48	0.21	0.80
T classification (Soft tissue tumor)				Retroperitoneum	0.82	0.81	0.99
T1	0.73	0.57	0.39	Local control	0.41	0.72	0.31
T2	0.52	0.36	0.85	Total dose	0.039	0.017	0.17
T3	0.16	0.51	0.70	GTV volume	0.68	0.11	0.78
Tumor grading				Base line score	<0.0001	<0.0001	<0.0001
G1	0.14	0.82	0.60	Late adverse event	0.07	0.79	0.63
G2	0.51	0.99	0.97	(Grade 2 or more)			
G3	0.41	0.93	0.11				
GX	0.19	0.80	0.23				

Abbreviations: MSTS = musculoskeletal tumor society score; PCS = physical component score; MCS = mental component score; Sarcoma, NOS = sarcoma not otherwise specified; MPNST = malignant peripheral sheath tumor; GTV = gross tumor volume.

**Table 3 cancers-13-02591-t003:** Predictive factors for the normalized scores determined using uni- and multivariate analyses.

**MSTS**						
**Characteristics**	**Univariate**			**Multivariate**		
	β	95% CI	*p*	β	95% CI	*p*
Local control	0.29	−0.4, 0.99	0.41	0.21	−0.34, 0.71	0.48
Adverse event	0.58	−0.049, 1.2	0.07	0.3	−0.15, 0.81	0.17
GTV volume	−0.00015	−0.0009, 0.00059	0.68	−0.00033	−0.00084, 0.00028	0.32
KPS	−0.055	−0.073, −0.037	<0.0001	−0.057	−0.070, −0.034	<0.0001
Sex	−0.73	−1.26, −0.19	0.0085	0.66	0.20, 1.1	0.0052
Total dose	0.12	0.0061, 0.23	0.039	0.00023	−0.10, 0.093	0.93
**PCS of SF−8**						
**Characteristics**	**Univariate**			**Multivariate**		
	β	95% CI	*p*	β	95% CI	*p*
Local control	0.048	−0.21, 0.31	0.72	0.037	−0.20, 0.28	0.75
Adverse event	0.031	−0.21, 0.27	0.79	−0.027	−0.24, 0.019	0.81
GTV volume	0.000066	−0.00021, 0.00034	0.63	−0.000029	−0.0024, 0.00028	0.82
KPS	−0.014	−0.022, −0.0059	0.0008	−0.013	−0.020, −0.0042	0.0037
Sex	−0.25	−0.46, −0.054	0.014	0.21	0.0043, 0.41	0.045
Total dose	0.047	0.0053, 0.088	0.028	0.014	−0.030, 0.057	0.54
**MCS of SF−8**						
**Characteristics**	**Univariate**			**Multivariate**		
	β	95% CI	*p*	β	95% CI	*p*
Local control	−0.069	−0.20, 0.064	0.31	−0.068	−0.20, 0.069	0.32
Adverse event	0.056	−0.15, 0.093	0.63	0.05	−0.14, 0.12	0.88
GTV volume	−0.00003	−0.00017, 0.0070	0.67	−0.000022	−0.000017, 0.00013	0.77
KPS	0.0026	−0.0017, 0.0070	0.23	0.0026	−0.0022, 0.0073	0.28
Sex	0.054	−0.054, 0.16	0.32	0.057	−0.062, 0.18	0.34
Total dose	0.0016	−0.021, 0.024	0.89	0.00062	−0.025, 0.026	0.96

The results of the univariate and multivariate logistic regression models are shown. In the “Sex” category, male was indicated as 0 and female was indicated as 1. Adverse events refer to late adverse events. Late adverse events classified as Grade 1 or less corresponded to 0, and those graded as Grade 2 or more corresponded to 1. Abbreviations: MSTS = musculoskeletal tumor society score; PCS = physical component score; MCS = mental component score; CI = confidence interval; *p* = *p*-value.

## Data Availability

The data are not publicly available due to their containing information that could compromise the privacy of research participants.

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
