# Peer review of "Prospective Evaluation of Quality of Life and Functional Outcomes after Carbon Ion Radiotherapy for Inoperable Bone and Soft Tissue Sarcomas"

_cancers, 2021, doi:10.3390/cancers13112591_

Round 1

Reviewer 1 Report

Nicely done overall. 

Hopefully you can update us in 3 or 4 years with more data. 

Author Response

We sincerely thank the reviewer for evaluating our manuscript. We will update our study in 3 or 4 years with longer follow-up data.

Reviewer 2 Report

Thank you for the opportunity to review this manuscript again. The authors have made some additions and changes to the manuscript, but I still think that this version can be improved significantly. The authors did not clarify why so many patients with small tumors or G1 sarcomas were classified as inoperable. It has been shown that there is a bias regarding the indication for radiation vs surgery in some countries, which is probably present but not discussed in this paper. The QL question should be at the forefront for all colleagues treating such a patient population, especially when it comes to non-curable tumors/patients, however I cannot see why so many patients did not receive a curative approach.
Here I would like to see much more information with the given conclusion.

Author Response

Reviewer #2

Thank you for the opportunity to review this manuscript again. The authors have made some additions and changes to the manuscript, but I still think that this version can be improved significantly.

Response

We sincerely thank the reviewer for evaluating our manuscript. We thoroughly revised the manuscript according to the comments, as described below.

The authors did not clarify why so many patients with small tumors or G1 sarcomas were classified as inoperable. It has been shown that there is a bias regarding the indication for radiation vs surgery in some countries, which is probably present but not discussed in this paper. The QL question should be at the forefront for all colleagues treating such a patient population, especially when it comes to non-curable tumors/patients, however I cannot see why so many patients did not receive a curative approach.

Response

Almost all of the G1 sarcomas included in this study were sacral chordomas. We summarized the characteristics of our cohort in Table 1. Our treatment policy is to give priority to carbon ion radiotherapy for chordomas that extend to S1, regardless of the size, considering the neurological damage such as bladder and rectal disorders that should occur after surgery. Other T1 bone tumors were considered unresectable, mainly because of the patient's advanced age or serious complications. All indications for CIRT are determined by a cancer review board that includes orthopedic surgeons, radiotherapists, and medical oncologists, and if surgical resection is obviously more beneficial, CIRT was not indicated and not performed.

We agree with your suggestion that we should discuss the bias deriving from deriving from decisions on inoperability. This point was added in the paragraph of the Discussion that describes the potential bias of the decisions on inoperability in our results. In brief, Decisions on tumor resectability and inoperability are generally made by a cancer review board, considering risks/benefits and availability of the medical options in the region. CIRT is currently an extremely limited medical resource. We should consider the potential bias deriving from decisions on inoperability in the individual situation with different medical options.

This manuscript is a resubmission of an earlier submission. The following is a list of the peer review reports and author responses from that submission.

Round 1

Reviewer 1 Report

  1. It is unclear that the entire analysis is valid. The MCS and KPS are treated as independent. They are not. The analysis in general has to be done on a per patient basis. It is unclear that was done. As such, the entire analysis of this paper should be done by a formal biostatistician and said person should be references as such - perhaps like translation was done, as a consultant. For example, if one patient went from very high to very low in a score, or 10%, could this analysis note if 10% went the opposite direction and therefore would balance out in the overall "average" score...? Given the various locations of the tumors and specific plans and issues, it is not clear that the approach used is valid or even appropriate. 
  2. These data need to be placed in a table and formally compared to recent/best literature by others. Discussion must discuss things in context. There is no context in the article. I am not clear why this is better than protons or even good surgery. Was unresectable reviewed by an outside panel at any time or could it be....were 4 soft tissue T1aN0M0 really unresectable?  
  3. Rhabdomyosarcoma should be treated with chemotherapy as could some of the others ...that is a confounder - did you not give patients standard of care chemo (i.e. for the rhabdomyosarcoma patient)  in this case and perhaps some of the other chemo sensitive histologies or did you fail to note that in the paper.
  4. What dose sex mean in terms of being signficant? How can it be studied given the lack of equal numbers and likely location and histlogy variation across sex?

Reviewer 2 Report

Thank you very much for the opportunity to review this really interesting paper.
It is an interesting paper on a timely topic.
Overall, the paper is well done, and well written.
However, some questions remain unanswered which should be clarified:
How was non-operable defined? For example, 10 of the bone tumors are smaller than 8 cm. ~ 75% of the patients have a KPS of 90 and more. It is interesting why tumors were classified as non-resectable.
Why was no (neo-) adjuvant chemotherapy used for inoperable tumors?
Why was only MSTS and SF-8 analyzed?
Are there any differences in the results regarding the scores if bone and soft tissue sarcomas are evaluated separately?
Are there localizations that are considered to benefit particularly compared to conventional photon irradiation?

There is little discussion of comparison with photon therapy, perhaps this can be added further.

Reviewer 3 Report

This study analyzed prospectively quality of life (SF8) and function (MSTS) after definitive carbon ion radiotherapy in 61 unresectable soft tissue sarcomas.

  • The study enrolled a heterogenous population in terms of tumor location and histology, factors that could affect the functional outcome and patient’s QoL.
  • Moreover, patient age is not mentioned in the study, I think that it is an important characteristic that has to be considered.
  • The SF-8 generates a health profile of eight scores describing quality of life, it summarized physical component (PCS) and mental component (MCS) . It is unclear why the authors use the SF8, nowadays, SF8 is not largely used to define quality of life and different scales are available in this field (van Roij J, Fransen H, van de Poll-Franse L, Zijlstra M, Raijmakers N. Measuring health-related quality of life in patients with advanced cancer: a systematic review of self-administered measurement instruments. Qual Life Res. 2018 Aug;27(8):1937-1955. doi: 10.1007/s11136-018-1809-4. Epub 2018 Feb 10. PMID: 29427216).
  • To complete the analysis I suggest to correlate function by grade of radiation morbidity

In the text some review are suggested as follows:

  • REFERENCES 3,4: Referred to target and immunotherapy in soft tissue sarcoma, not to standard treatment (chemotherapy and radiotherapy actually indicated in unresectable soft tissue sarcoma, on this basis the limited therapeutic efficacy of these strategies cannot be affirmed) ( 47)
  • CIRT, an innovative radiotherapy modality that achieves high dose 50 conformity to deeply located tumors, is more effective than photon therapy : more effective in what? Local control, pathological response? Survival? (51)
  • Dose constraints for bone are not considered, the risk of bone fracture is high at the mentioned prescription doses (97)
  • In the table: please use the AJCC staging system ed. 2017
  • Please correct in the table: Soft tissue tumor in T classification
  • Tumor grading, it could be useful to correlate to clinical outcome
  • Please specify the time when late toxicities were recorded, 90-day rule is often used. Fibrosis can develop over a 2-year period.

Reviewer 4 Report

The work presented by Komatsu et al. focus on the role of quality of life and functional outcomes in patients with unresectable bone and soft tissue sarcoma treated with carbon-ion radiotherapy (CIRT). They analyzed prospectively 61 patients affected by unresectable bone and soft tissue sarcoma who were treated with CIRT. Muscoloskeletal Tumor Society Score (MSTS) and physical component score (PCS) were assessed. Multivariate analysis showed that normalized MSTS and normalized PCS of Short Form questionnaire (SF-8) were significalntly affected by by performance status and at diagnosis and sex. Moreover the results highlighted the clinical efficacy of CIRT, preserving the physical and mental component of quality of life of patients with unresectable bone and soft tissue sarcoma. The results pointed out the promising value of CIRT for these patients.

Minor concerns

  1. Diagnostic features of the analyzed case series should be implemented. No data about histopathology are provided (for instance, dedifferentiated, well differentiated, myxoid and pleomorphic liposarcoma. MDM2 amplification, FUS-CHOP translocation). In this regard the work titled “Current classification, treatment options, and new perspectives in the management of adipocytic sarcomas” Onco Targets Ther. 2016 Oct 11;9:6233-6246. doi: 10.2147/OTT.S112580 should be included in the references.
  2. In the results section the authors should delete the sentence of the journal format, “this section may be divided...”.
  3. Limitation of the study should be discussed

Minor corrections are required before publication